# Chain-of-Thought Reasoning without Prompting

**Xuezhi Wang**
Google DeepMind
xuezhiw@google.com

**Denny Zhou**
Google DeepMind
dennyzhou@google.com

## Abstract

In enhancing the reasoning capabilities of large language models (LLMs), prior research primarily focuses on specific prompting techniques such as few-shot or zero-shot chain-of-thought (CoT) prompting. These methods, while effective, often involve manually intensive prompt engineering. Our study takes a novel approach by asking: Can LLMs reason effectively without prompting? Our findings reveal that, intriguingly, CoT reasoning paths can be elicited from pre-trained LLMs by simply altering the *decoding* process. Rather than conventional greedy decoding, we investigate the top-$k$ alternative tokens, uncovering that CoT paths are frequently inherent in these sequences. This approach not only bypasses the confounders of prompting but also allows us to assess the LLMs' *intrinsic* reasoning abilities. Moreover, we observe that the presence of a CoT in the decoding path correlates with a higher confidence in the model's decoded answer. This confidence metric effectively differentiates between CoT and non-CoT paths. Extensive empirical studies on various reasoning benchmarks show that the proposed CoT-decoding effectively elicits reasoning capabilities from language models, which were previously obscured by standard greedy decoding.

## 1 Introduction

Large language models (LLMs) have demonstrated remarkable performance on various complicated reasoning benchmarks (Anil et al., 2023; Brown et al., 2020; Chowdhery et al., 2023; Gemini, 2023; OpenAI, 2023; Romera-Paredes et al., 2023). These reasoning capabilities of LLMs are typically elicited by prompting techniques (Brown et al., 2020), which can be few-shot prompting with intermediate steps augmented demonstration exemplars (Chen et al., 2023b; Gao et al., 2022; Nye et al., 2021; Wei et al., 2022; Yao et al., 2023; Zhou et al., 2023a), or zero-shot prompting with specific instructions which ask for showing certain intermediate steps (Kojima et al., 2022; Yasunaga et al., 2023). The other prevalent strategy for eliciting LLM reasoning is through model training or instruction tuning using a substantial amount of chain-of-thought (CoT) reasoning data (Chung et al., 2022; Cobbe et al., 2021; Ling et al., 2017; Nye et al., 2021).

Prompting techniques, while effective, often encode task-specific human priors, thereby making it difficult to assess a language model's intrinsic reasoning abilities. Ideally, a well-trained language model should be capable of independent reasoning and delivering optimal responses, without requiring humans to tweak the prompts or refine repeatedly when the initial response is unsatisfactory. Model-tuning can be expensive and requires a substantial amount of supervised data. In this work, we explore a different perspective and ask: *Can LLMs reason effectively without prompting? And to what extent can they reason?* We find that, perhaps surprisingly, there exists a *task-agnostic* way to elicit CoT reasoning from pre-trained LLMs by simply altering the *decoding* procedure. Figure 1 illustrates this phenomenon: given a reasoning question, the LLM generates a wrong answer via the standard greedy decoding path, yet alternative top-$k$ token inspection unveiled inherent CoT paths (e.g., decoding paths 2 and 4), which accurately resolved the query. This decoding modification bypasses prompting and is entirely unsupervised without the need for model tuning.

38th Conference on Neural Information Processing Systems (NeurIPS 2024).

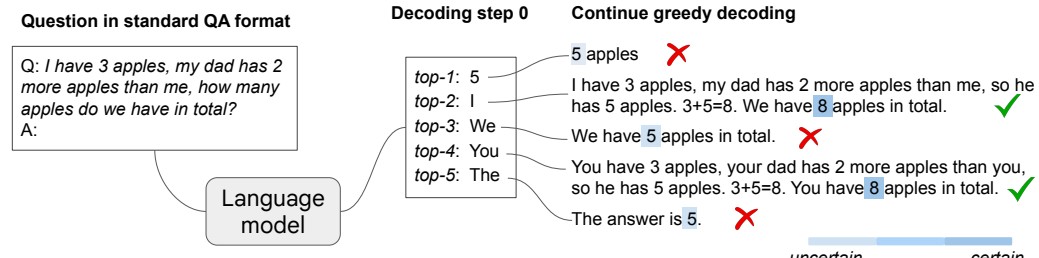

Figure 1: **Illustration of CoT-decoding**. Pre-trained LLMs are capable of inherent reasoning without prompting by considering alternative top-$k$ tokens, rather than solely relying on the top-1 greedy decoding path. Moreover, these models tend to display higher confidence in decoding the final answer (indicated by a darker shaded color) when a CoT reasoning path is present.

In more details, we formulate the input using the standard question-answer (QA) format: "Q: [question]\nA:".[1] While most existing work suggest that LLMs falter in such direct-QA scenarios on reasoning (Cobbe et al., 2021; Kojima et al., 2022; Nye et al., 2021; Wei et al., 2022), our findings reveal a nuanced picture. We observe that LLMs indeed struggle with reasoning when relying solely on greedily decoded paths. However, when we consider alternative paths among the top-$k$ tokens, CoT reasoning patterns *emerge naturally* within the decoding trajectories of LLMs. In addition, we have observed an interesting pattern: the model demonstrates increased confidence in the final answer when a CoT reasoning path is present in the decoding process. As illustrated in Figure 1, this is evident where paths 2 and 4 show heightened certainty in arriving at the correct answer "8", contrasting sharply with the high uncertainty in paths that lead to the incorrect "5". Leveraging this phenomenon, we develop a method to sift through the top-$k$ decoding paths, which we refer to as **CoT-decoding**, thereby isolating the most reliable paths for model output.

Our contributions are summarized as follows:

- **We present a novel finding that LLMs can reason by simple decoding changes, without the use of prompting**. In contrast to prior research that focuses on refining prompts to elicit reasoning from LLMs, our work, for the first time, shows that the reasoning process can be readily elicited by simple decoding changes. Moreover, we challenge the prevailing notion in the literature that LLMs are inherently incapable of effective reasoning without prompting. We show that this belief is an artifact of considering only the greedy path during decoding, and the model's reasoning paths can be revealed by traversing the alternative decoding paths.

- **Our method enables a better understanding of LLMs' intrinsic reasoning capabilities without imposing human priors**. The employment of intricate prompting techniques often introduces various human priors, making it difficult to distinguish between the extent of "human teaching" and the degree to which LLMs can reason independently. Our approach bypasses the confounders introduced by prompting, enabling a more truthful assessment of the models' intrinsic reasoning abilities. Our study reveals that pre-trained language models *inherently possess reasoning capabilities* for many tasks including math and commonsense reasoning, and existing prompting approaches mostly serve the role of bringing those inherent reasoning paths forward as the top decoding paths. In contrast, the CoT-paths are less prevalent in complex and highly synthetic tasks, where the few-shot CoT demonstrations play a "teaching" role in guiding how models solve a task, with models primarily mimicking the format of these prompts to generate accurate reasoning paths.

- **We further propose CoT-decoding that reliably selects CoT-paths based on answer confidence.** We find that the language model's confidence in its final answers increases when a CoT is present in its decoding path. Leveraging this increased confidence, we propose **CoT-decoding** to select more reliable decoding paths, demonstrating significant improvements over greedy decoding across various reasoning benchmarks.

---

[1]The QA format is only needed because without it a pre-trained language model will continue the question instead of answering. It is also the most basic formatting employed in existing works for pre-trained models.

## 2 Chain-of-Thought (CoT) Decoding

### 2.1 Pre-trained Language Models Can Reason without Prompting

We investigate whether pre-trained language models inherently possess reasoning capabilities, without explicit prompts or human intervention. In Table 1, we show example decoding paths across math (GSM8K, Cobbe et al. (2021)) and commonsense reasoning (year parity, Allen-Zhu and Li (2023)). We employ the *pre-trained* PaLM-2 large model (Anil et al., 2023) to compare its greedy decoding path ($k = 0$), predominantly used in state-of-the-art LLMs for reasoning tasks, with alternative decoding paths ($k > 0$), where $k$ represents the choice of the $k$-th token at the *first decoding step*.

| **[GSM8K]** *One glass costs $5, but every second glass costs only 60% of the price. Kylar wants to buy 16 glasses. How much does he need to pay for them?* | **[Year Parity]** *Was Nicolas Cage born in an even or odd year?* |
|---|---|
| **Greedy path**: 
 $k = 0$: **$60.00** (0.029) 
 **Alternative top-$k$ paths**: 
 $k = 1$: **60** (0.058) 
 $k = 2$: Kylar needs to pay **$60** for 16 glasses. (0.058) 
 . . . 
 $k = 7$: If Kylar buys 16 glasses, he will pay **$60**. (0.032) 
 $k = 9$: We can calculate . . . we need to multiply the price of one glass by 16 and then subtract 40% of the price of 8 glasses. 16 x 5 = 80 8 x 5 = 40 40 x 0.4 = 16 80 – 16 = 64 Kylar needs to pay **$64** for 16 glasses. (0.994) | **Greedy path**: 
 $k = 0$: Nicolas Cage was born in an **odd** year. (0.117) 

 **Alternative top-$k$ paths**: 
 $k = 1$: **Even** (0.207) 
 $k = 2$: **Odd** (0.198) 
 $k = 3$: 1964, an **even** year. (0.949) 
 $k = 4$: He was born in an **even** year. (0.0) 
 . . . 
 $k = 7$: Cage was born in 1964, an **even** year. (0.978) |

Table 1: Examples of greedy decoded paths and alternative top-$k$ paths over the PaLM-2 Large model. The model's confidence over the answers (bolded) are highlighted in blue (See §2.2 for details).

**LLMs indeed cannot reason if we only consider the greedy decoding path.**   First, we observe that models employing greedy decoding often does not contain a CoT path, opting to solve problems directly. This tendency may stem from the model's skewed perception of problem difficulty, shaped by its pre-training on predominantly simpler questions. Consequently, the model is predisposed to immediate problem-solving. This observation aligns with findings in (Cobbe et al., 2021; Kojima et al., 2022; Nye et al., 2021; Wei et al., 2022), which show that direct-answer prompts generally result in low accuracy on reasoning tasks even for large language models.

**LLMs can reason if we consider the alternative decoding paths.**   Contrastingly, an intriguing phenomenon emerges when exploring alternative top-$k$ ($k > 0$) tokens at the *first decoding step*. Continuing with greedy decoding from this point reveals natural CoT reasoning in many cases. These findings suggest that large language models possess inherent reasoning capabilities for numerous tasks following pre-training, but these abilities are obscured by the predominant use of greedy decoding. These reasoning paths can be easily uncovered by incorporating alternative decoding paths.

For instance, in the GSM8K question (Table 1), a valid CoT emerges at $k = 9$. Similarly, in the year parity task, greedy decoding attempts to directly answer the parity question at $k = 0$, leading to a random choice between "even" and "odd" which often results in an incorrect answer. However, when exploring $k > 0$, the model naturally generates CoT paths at $k = 3$ and $k = 7$, where it first determines the year before resolving the parity.

### 2.2 CoT-Decoding for Extracting CoT Paths

In this section, we further show how we can reliably extract those CoT-paths during the decoding process. We observe that CoT paths do not consistently outrank non-CoT ones in the model's probability assessment. Moreover, they often do not represent the predominant answer among all paths, rendering methods like self-consistency (Wang et al., 2023a) inapplicable. For instance, in the GSM8K question, the prevalent answer "60", which aligns with the greedy decoding result, fails to serve as a reliable indicator for identifying the correct path.

Interestingly, upon examining the model's logits, we found that the presence of a CoT path typically leads to a more *confident* decoding of the *final answer*, characterized by a significant probability disparity between the top and secondary tokens:

$$\Delta_{k,\text{answer}} = \frac{1}{|\text{answer}|} \sum_{x_t \in \text{answer}} p(x_t^1 \mid x_{<t}) - p(x_t^2 \mid x_{<t}).$$

Here $x_t^1$ and $x_t^2$ represent the top two tokens at the $t$-th decoding step in the $k$-th decoding path, chosen for their maximum post-softmax probabilities from the vocabulary, given $x_t$ being part of the answer tokens. This uncertainty measure is similar to the minimum-margin approach in (Jiang and Gupta, 2019) and in our case, the model's overall confidence in decoding the final answer is approximated by averaging these probability differences for all relevant answer tokens $x_t$. For example, for the GSM8K question in Table 1, given the answer "60", we average the probability differences for all tokens in that answer, i.e., "6" and "0".[2]

This method, referred to as **CoT-decoding**, extracts such CoT paths among the decoded paths from the model. As illustrated in Table 1, each decoding path is marked with its corresponding $\Delta$ value in blue (the answer tokens are bolded). It is evident that paths with a CoT component exhibit a significantly higher $\Delta$, highlighting the model's increased confidence, as opposed to paths without CoT. We also did a quantitative analysis by manually examining the first 100 questions in GSM8K, and among those, if we take the decoding path with the highest answer confidence among the top-10 decoding paths, 88% of them contain CoT paths. This shows an overwhelmingly high correlation between the model's answer confidence and the CoT paths.

**Comparing different CoT-path extraction approaches.** In Table 2, we compare different ways to extract the CoT-paths out of the top-10 decoded paths. It is easy to see that the model's own probability measure does not serve as a reliable indicator, nor does the model's length-normalized probability (since an intuition could be a CoT-path should usually be a longer decoding path, which is not always the case, e.g., on the year parity task). In contrast, CoT-decoding can reliably extract the CoT-paths, yielding a significant boost on the model's reasoning performance.

|  | GSM8K (top-100) | Year Parity |
|---|---|---|
| Greedy decoding | 44.0% | 57.0% |
| Decode 10 paths, rank by model's highest log-prob | 37.0% | 55.0% |
| Decode 10 paths, rank by model's highest length-normalized log-prob | 51.0% | 57.0% |
| CoT-decoding (decode 10 paths, rank by model's answer confidence) | **72.0%** | **95.0%** |

Table 2: CoT-decoding reliably extracts the CoT-paths compared to other methods (on PaLM-2 L).

**Identify the answer spans.** Computing $\Delta$ requires identifying the answer spans in a model's response. One common approach used for public models is to extract the last numerical value in math reasoning tasks, or the final option in set-based reasoning tasks, as the answer, following the Tülu evaluation (Ivison et al., 2023; Liu et al., 2024; Wang et al., 2023b). Alternatively, similarly to the method used in Kojima et al. (2022), we can extend the model's output with the prompt "So the answer is", and then align these continuations with spans in the model's decoding path as the answer.

**Sampling under the standard QA format.** CoT-decoding explores alternative tokens at the first decoding step. A natural question arises: can sampling achieve a similar effect and unveil the CoT reasoning paths? We

Table 3: CoT-decoding and self-consistency w/o prompts on GSM8K.

|  | Mistral-7B | PaLM-2 L |
|---|---|---|
| Greedy decoding | 9.9% | 34.8% |
| Self-consistency without CoT-prompt (10 paths) | 12.9% | 40.6% |
| CoT-decoding (10 paths) | **25.1%** | **63.2%** |

found that, although sampling works well under few-shot CoT prompting (Wang et al., 2023a), it does not exhibit the desired behaviour without the prompts. In Table 3, we show that when no CoT prompt is used, CoT-decoding is much more sample-efficient than sampling-based approaches like self-consistency. The ineffectiveness of sampling stems from the model's skewed distribution towards the direct answer tokens, hence the first token tends to have less diversity compared to CoT-decoding. In contrast, CoT-decoding works by explicitly encouraging diversity at the first decoding step.

**Branching at other decoding steps.** Another natural question is whether branching is viable at later decoding stages, comparing to only branching at the first decoding step. In Figure 2, we highlight

---

[2]We also considered other popular choices for measuring the model's uncertainty (Settles, 2009), e.g., using the model's probability on the token itself (i.e., $p(x_t^1 \mid x_{<t})$ only), which performs slightly worse compared to the min-margin approach. In addition, an entropy estimate is not accurate due to the large vocabulary size in LLMs and the common use of vocabulary truncation.

the impact of alternative token consideration in subsequent decoding steps. It is evident that early branching, e.g., at the first decoding step, significantly enhances the diversity of potential paths. Conversely, later-stage branching is significantly influenced by previously generated tokens. For instance, initiating with the token "5" greatly decreases the likelihood of rectifying an erroneous path. Nonetheless, the optimal branching point may vary with the task; in the year parity task, for instance, mid-path branching can effectively yield correct CoT paths.

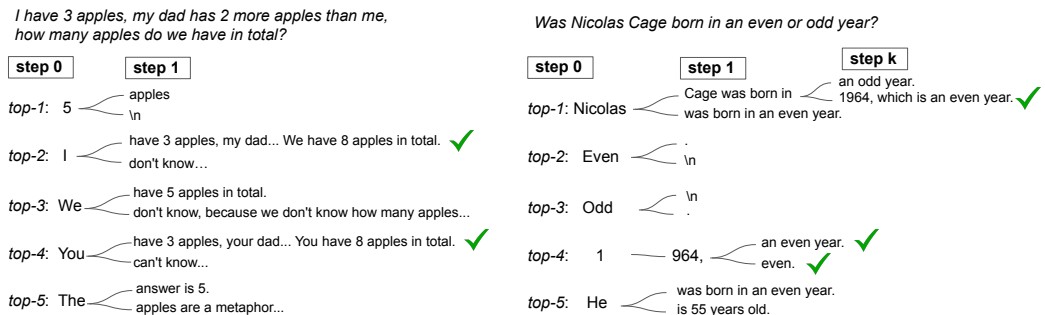

Figure 2: We present an analysis of the decoded paths by considering alternative tokens at various decoding steps. Task-dependent challenges arise: at times, the model encounters difficulty recovering from incorrect paths when branching at later tokens. For certain tasks, multiple branching positions may exist, all leading to the correct reasoning path.

**Aggregation of the decoding paths.** Since we already decode the top-$k$ paths, one natural extension is to aggregate the answers over all those paths, similar to self-consistency (Wang et al., 2023a) but without the use of prompts. The rationale behind this aggregation is to mitigate sensitivity to small differences in the model's logits, particularly when relying solely on the path with the maximum $\Delta$. The examples in Table 1 show that the majority answer is unlikely to be the correct one. Instead, we propose a weighted aggregation method, i.e., we take the answer that maximizes $\tilde{\Delta}_a = \sum_k \Delta_{k,a}$ where $\Delta_{k,a}$ is the $k$-th decoding path whose answer $= a$. We found that adopting this approach enhances the stability of the results, and further analysis is presented in Section §3.3.

## 3 Experiments

We evaluate CoT-decoding across a range of reasoning benchmarks, demonstrating its capability to successfully recover CoT reasoning paths during decoding, all without the need for specialized prompting.

**Experiment Setup.** For all experiments, the default input to the model is the standard QA format of *Q: [question]\nA:*, where *[question]* is filled with the actual question depending on the task, and we ask the model to continue the generation given that prefix. During decoding, we use $k = 10$ as default for the alternative top-$k$ tokens at the first decoding position, and continue greedy decoding afterwards. We show ablation studies with respect to the different choice of $k$ in Section §3.1.

**Datasets.** For mathematical reasoning, we use the Grade-school math problems (GSM8K; Cobbe et al., 2021) and the multi-step arithmetic dataset from (MultiArith; Roy and Roth, 2015). For commonsense reasoning, we investigate the "year parity" task where recent literature finds large language models still struggle with. The task is to query the model with "Was [person] born in an even or odd year?" where "[person]" is filled by a random celebrity name.[3] Existing work (Allen-Zhu and Li, 2023; Berglund et al., 2023) shows that even SoTA models like GPT-4 struggle with such tasks, achieving at-chance accuracy (∼50%) when prompted directly. Additionally, we investigate symbolic reasoning tasks from Big-Bench-Hard (bench authors, 2023; Suzgun et al., 2022).

**Models.** We use three public models: (1) PaLM-2 (Anil et al., 2023) with different scales, ranging from X-Small, Small, Medium, and Large; (2) Mistral-7B (Jiang et al., 2023), and (3) Gemma-7B

---

[3]We curate a list of the top 100 celebrity names from (Berglund et al., 2023): `https://github.com/lukasberglund/reversal_curse/blob/main/data/celebrity_relations/top_celebrities.txt`

(Team et al., 2024). Our experiments primarily focus on pre-trained models, but we also include experiments with instruction-tuned models (denoted as "inst-tuned" or "IT").

## 3.1 CoT-Decoding Effectively Elicits Reasoning from Language Models

**CoT-decoding is the only decoding strategy that effectively elicits language model reasoning.** In Table 4, we present results from popular decoding baselines on the Mistral-7B pre-trained model, including temperature sampling (Ackley et al., 1985; Ficler and Goldberg, 2017), top-$k$ sampling (Fan et al., 2018; Holtzman et al., 2018; Radford et al., 2019), nucleus sampling (Holtzman et al., 2020), and beam search. We can see that CoT-decoding is the only decoding strategy that effectively

Table 4: CoT-decoding is the only decoding strategy that can effectively elicit language models' reasoning.

|  | GSM8K Acc |
|---|---|
| Top-$k$ sampling ($k = 10$) | 4.9% |
| Top-$p$ / Nucleus sampling ($p = 0.9$) | 6.4% |
| Beam search ($b = 10$) | 6.7% |
| Temperature sampling ($T = 0.7$) | 7.5% |
| Greedy decoding | 9.9% |
| Self-consistency w/o CoT prompt (10 paths) | 12.9% |
| CoT-decoding ($k = 10$) | **25.1%** |

enables language models to reason, while some of the decoding methods even hurt model's reasoning ability compared to greedy decoding.

**CoT-decoding effectively elicits reasoning across language models.** In Figure 3, we show that across three language model families, PaLM-2, Mistral and Gemma, CoT-decoding yields consistent accuracy gains on both math and commonsense reasoning tasks, sometimes doubling or even tripling the performance compared to greedy decoding.

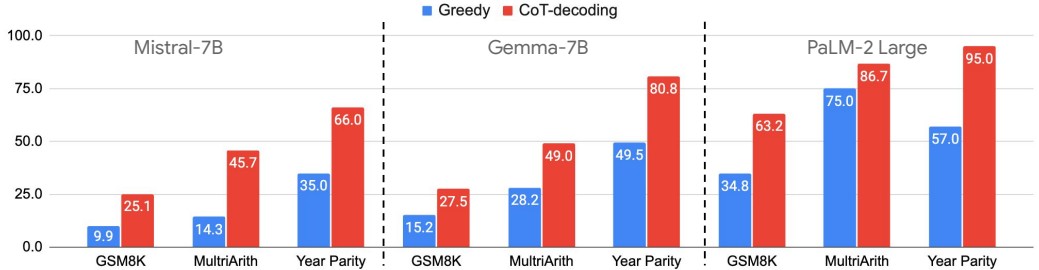

Figure 3: CoT-decoding effectively elicits reasoning across multiple language model families including PaLM-2, Mistral and Gemma, with significant accuracy gains over three reasoning tasks.

**CoT-decoding elicits reasoning across model scales.** In Figure 4, we show that CoT-decoding enhances reasoning across different model scales over the PaLM-2 model family. On GSM8K, CoT-decoding consistently yields +10-30% absolute accuracy gains. On year parity, when using greedy decoding, the model's performance remains flat even after scaling up model sizes, consistent with the findings in (Allen-Zhu and Li, 2023). In contrast, CoT-decoding significantly boosts the performance by recovering the CoT paths, achieving almost perfect accuracy at larger model scales.

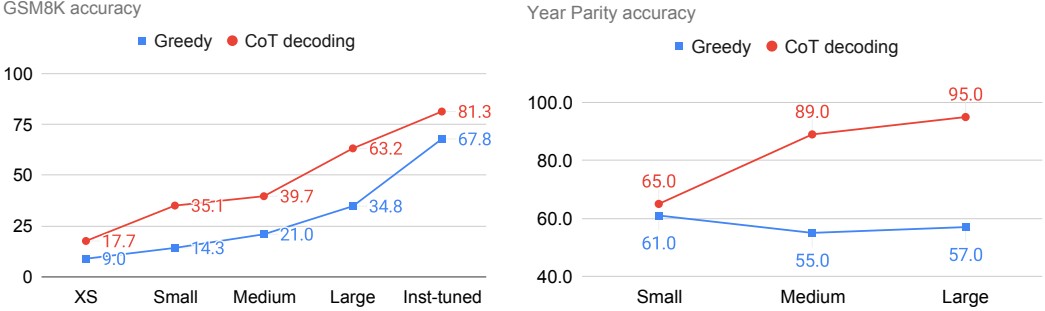

Figure 4: CoT-decoding reliably improves reasoning performance across model scales (PaLM-2), even when the task does not naturally improve by scaling up only (e.g., year parity).

**CoT-decoding partially closes the reasoning gap between pre-trained and instruction-tuned models, without using any supervised data.** Intriguingly, we observe that CoT-decoding enables a pre-trained model to achieve a similar performance of an instruction-tuned model: in Figure 4 (left), CoT-decoding achieves 63.2% accuracy on the pre-trained PaLM-2 Large model, close to the performance of the instruction-tuned model of the same scale at 67.8%. The results demonstrate that instruction-tuning with sufficient CoT data (Chung et al., 2022) can be partially achieved by modifying the decoding procedure within pre-trained models.

More interestingly, we observe that CoT-decoding can further improve the instruction-tuned model (Figure 4 (left) and Table 5). The instruction-tuning procedure (Chung et al., 2022) incorporated a large amount of CoT annotations during the fine-tuning process. Consequently, the model is expected to inherently generate CoT paths when addressing reasoning tasks. However, upon analyzing specific examples, we found that even after instruction-tuning, the model occasionally persists in attempting to directly address a question. In contrast, CoT-decoding can enhance the exploration of alternative paths by triggering a CoT first, consequently leading to more accurate answers.

Table 5: CoT-decoding improves both pre-trained and instruction-tuned Mistral-7B models.

|  |  | Pre-trained | Inst-tuned |
|---|---|---|---|
| GSM8K | Greedy | 9.9 | 31.2 |
|  | CoT-decoding | **25.1** (+15.2) | **38.2** (+7.0) |
| MultiArith | Greedy | 14.3 | 37.8 |
|  | CoT-decoding | **45.7** (+31.4) | **66.5** (+28.7) |
| Year Parity | Greedy | 35.0 | 62.2 |
|  | CoT-decoding | **66.0** (+31.0) | **73.5** (+11.3) |

**Choice of $k$.** In Figure 5, we illustrate how the choice of $k$, representing the number of top alternative tokens considered, influences the overall accuracy. Overall we found that higher values of $k$ typically result in improved model performance, suggesting that in many cases, the correct CoT paths may indeed exist but are often ranked lower during model's decoding. For instruction-tuned models, the effect of $k$ is less significant, indicating that the process of instruction-tuning effectively brings forth the majority of CoT-paths to the first few decoding paths.

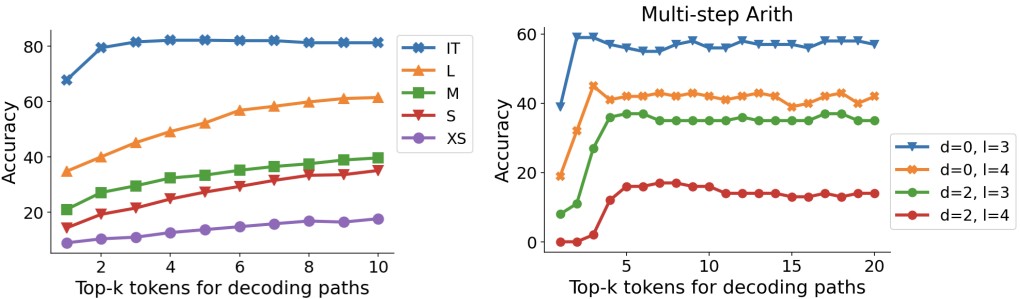

Figure 5: The effect of $k$ on reasoning accuracy w.r.t. PaLM-2 model scales and task difficulty.

## 3.2 CoT-decoding Enables a Better Understanding of Model's Intrinsic Reasoning Abilities

Compared to existing works that improve model's reasoning via better human-written prompts, a key distinction of our proposed approach lies in the complete elimination of human-provided prompts. This modification enables a *more truthful assessment* of a language model's intrinsic reasoning capabilities. In the previous section, we show that language models inherently possess reasoning capabilities for grade-school-level math problems and simple commonsense reasoning tasks. In this section, we will systematically vary the difficulty levels of synthetic tasks to gain a more comprehensive understanding of language models' inherent reasoning abilities via CoT-decoding.

We consider the following symbolic reasoning tasks: (1) the Coin Flip task from (Wei et al., 2022), with $2, 3, 4$ rounds of potential flips; and two tasks from Big-Bench-Hard (bench authors, 2023; Suzgun et al., 2022): (2) Web of lies, with $3, 4, 5$ truth/lie statements, and (3) Multi-step arithmetic with various depth level $d$ and length $l$. For each task, we produce 100 examples for each difficulty level, except for Web-of-Lies (5) we use the existing dataset from (Suzgun et al., 2022). We also

include two natural-language-based but synthetic tasks from Big-Bench, Sports Understanding and Object Counting, to probe model's intrinsic abilities in solving synthetic tasks.

| | Coin Flip | | | Web of lies | | | Multi-step Arithmetic | | | | Sports Und. | Object Count |
|---|---|---|---|---|---|---|---|---|---|---|---|---|
| | 2 | 3 | 4 | 3 | 4 | 5 | $d_0, l_3$ | $d_0, l_4$ | $d_2, l_3$ | $d_2, l_4$ | | |
| Greedy | 70.0 | 53.0 | 48.0 | 76.0 | 58.0 | 53.6 | 39.0 | 19.0 | 8.0 | 0.0 | 58.8 | 36.0 |
| CoT-decoding | **94.0** | **57.0** | **55.0** | **87.0** | **63.0** | **57.6** | **56.0** | **42.0** | **35.0** | **16.0** | 58.0 | **39.2** |

Table 6: The model's intrinsic reasoning ability varies depending on the task difficulty levels.

**The presence of correct CoT paths depends on the task difficulty levels and correlates with task prominence in the pre-training distribution.** The results in Table 6 (on PaLM-2 L) show that despite CoT-decoding helps elicit better reasoning across almost all tasks, the gains vary significantly depending on the task difficulty level: the simpler the task is, the better chance that a correct reasoning path can be found. We also looked at the model's top-$k$ decoding paths, and found that models can generate the correct CoT paths when the solution involves at most 1 or 2 step knowledge manipulation, and the model starts to struggle with generating the correct CoT-paths when the steps become 3 or more. See Figure 5 (right) where the model's accuracy improves only for larger $k$'s as task complexity increases (higher $d$ and $l$'s). This phenomenon suggests that the correct CoT-paths become harder to find when the task becomes more synthetic. This mirrors the finding in (McCoy et al., 2023), where the authors show language models are highly influenced by the distribution they have been trained on.

**CoT-decoding unveils model's intrinsic vulnerabilities in reasoning.** Our results also unveil the specific areas where language models still struggle with: for example, on Coin-Flip and Web-of-Lies, we observe that the model can generate CoT paths that simulate the process step-by-step, but it can easily lose track of the states, especially when the task complexity increases. This reveals model's intrinsic vulnerability in performing accurate state tracking. On Multi-step Arithmetic, we observe that the model tends to perform calculations from left to right in the CoT-decoding paths, rather than following the correct mathematical order. These observations point to future directions where we should improve the models on.

In addition, over these synthetic tasks, we found that existing CoT prompts on Big-Bench-Hard (Suzgun et al., 2022) play a larger "teaching" role in guiding the model to solve such tasks, and in most cases the model just mimics the patterns in the CoT prompts to generate the correct response: e.g., the few-shot CoT prompts teach the model to perform *explicit* state tracking in each step for Web-of-lies. On the Sports Understanding task, CoT-decoding can better reveal LLMs' intrinsic strategy in solving a problem (see Appendix A), without being influenced by the external prompts which could be biased by the prompt designers. In contrast, few-shot CoT prompting constrains the model to follow an artificial strategy curated through human knowledge and intervention.

### 3.3   Combining CoT-decoding with CoT-Prompting

| | | Mistral-7B | PaLM-2 L | Compute |
|---|---|---|---|---|
| Methods without prompting | Greedy decoding | 9.9% | 34.8% | $\mathcal{O}(1)$ |
| | Self-consistency without CoT | 12.9% | 40.6% | $\mathcal{O}(k)$ |
| | CoT-decoding (max path) | 25.1% | 63.2% | $\mathcal{O}(k)$ |
| | CoT-decoding (agg path) | **25.3%** | **64.1%** | $\mathcal{O}(k)$ |
| Methods with prompting | Zero-shot CoT prompting | 17.5% | 75.1% | $\mathcal{O}(1)$ |
| | Self-consistency with zero-shot CoT-prompt | 39.4% | 85.3% | $\mathcal{O}(k)$ |
| | CoT-decoding (max path) + zero-shot CoT-prompt | 40.2% | 78.6% | $\mathcal{O}(k)$ |
| | CoT-decoding (agg path) + zero-shot CoT-prompt | **48.4%** | **87.0%** | $\mathcal{O}(k)$ |

Table 7: Adding CoT-decoding on top of zero-shot CoT-prompting can further boost the reasoning performance on both models. The accuracy number here is computed over the GSM8K test set.

We further show that CoT-decoding can be easily combined with CoT-prompting, yielding even larger reasoning gains over multiple language models (Table 7). CoT-decoding maintains a strong

performance compared to self-consistency (Wang et al., 2023a) when both are combined with CoT-prompts. Since self-consistency aggregates over multiple paths, we also show the performance based on our path aggregation algorithm, which significantly improves the model's reasoning at a similar cost. For a fair comparison, we use $k = 10$ for all methods that require multiple decoding paths.

## 4 Related Work

**Chain-of-thought reasoning in large language models.** Existing work enhancing the reasoning abilities in large language models predominantly involve proposing better prompting techniques to better elicit CoT reasoning paths from the model (Kojima et al., 2022; Nye et al., 2021; Wei et al., 2022; Yao et al., 2023; Yasunaga et al., 2023; Zhou et al., 2023a). Despite achieving high performance, few-shot prompting techniques are often *task-specific*, requiring prompt designs tailored to each task. This limits their generalizability across tasks. Advanced prompting techniques often require manually intensive prompt engineering, and their effectiveness varies depending on the choice of prompts, resulting in inconsistent performance outcomes (Wang et al., 2022; Ye and Durrett, 2022; Zhou et al., 2023b). Efforts to discover improved prompts (Yang et al., 2024; Zhou et al., 2023b) further entail model-specific and task-specific tuning.

In addition, these prompting techniques can subtly alter the vocabulary's posterior distribution in ways that remain largely elusive (Min et al., 2022; Webson and Pavlick, 2022). Specifically, prompts may assist in task decomposition, induce the model to generate additional tokens, or directly "teach" the model the exact underlying procedure to solve particular problems via manually crafted few-shot demonstrations. Dissecting the distinct influence of each aspect, however, presents a significant challenge. In contrast, our work explores a different perspective within the decoding stage, demonstrating that, even without explicit prompting, the model inherently holds the capability to generate chain-of-thought reasoning paths across a wide set of tasks.

Recent work proposes to improve the CoT generation process via better controlling and verifying the steps generated, e.g., step-by-step verification (Lightman et al., 2023), process-based feedback (Uesato et al., 2022), self-evaluation guided beam search (Xie et al., 2023), and PathFinder (Golovneva et al., 2023). Note all these works still require CoT prompting in order to generate the CoT reasoning paths, while our work completely removes CoT prompting. In addition, these works focus on searching and verifying the "steps" produced by the language model, while our work purely searches in the decoding space on the token-level and utilizes the confidence scores when decoding the answer.

Additionally, recent works (Feng et al., 2023; Li et al., 2023b; Prystawski et al., 2023). McCoy et al. (2023); Razeghi et al. (2022) demonstrate a similar phenomenon where the pretraining distribution heavily influences the model's performance in few-shot reasoning.

**Instruction-tuning to elicit CoTs in language models.** When supervision is allowed, techniques such as instruction-tuning or distillation offer another way to elicit reasoning paths from language models without explicit prompting (Chung et al., 2022; Huang et al., 2023; Magister et al., 2023). However, these approaches typically involve resource-intensive fine-tuning over large language models and require a large set of examples annotated with CoTs, which may not be readily available.

Liu et al. (2024) show that a language model can be tuned by a proxy. Their method requires a few additional models, and implicitly assumes that the tuned model is well-optimized, e.g., on reasoning benchmarks the model needs to be tuned with CoT paths to enable contrasting logits with respect to the base untuned model. In contrast, our approach is entirely unsupervised and examines a model's intrinsic ability in generating CoT paths, without resorting to fine-tuning or any additional models.

**Decoding algorithms for language models.** The predominant focus in existing literature on decoding for language models revolves around aspects such as fluency, coherence, reduction of repetitiveness, and diversity in responses. Popular decoding algorithms used for language models include greedy decoding, temperature sampling (Ackley et al., 1985; Ficler and Goldberg, 2017), top-$k$ sampling (Fan et al., 2018; Holtzman et al., 2018; Radford et al., 2019), and nucleus sampling (Holtzman et al., 2020). Additionally, there exist refined algorithms such as minimum Bayes risk decoding (Eikema and Aziz, 2020), and typical decoding (Meister et al., 2022). Diverse beam search (Vijayakumar et al., 2018) is another way to explore alternative paths in a model's generation. However, it emphasizes generation diversity rather than accuracy.

There is relatively little research dedicated to enhancing decoding algorithms specifically for reasoning tasks. Wang et al. (2023a) improves upon CoT prompting by sampling and aggregating over multiple generated responses to improve reasoning. Contrastive decoding (Li et al., 2023a) is another way to improve model's generation quality by penalizing the logits from smaller models, and recent work (O'Brien and Lewis, 2023) shows that contrastive decoding can contribute to enhancing reasoning performance. Shi et al. (2023) propose context-aware decoding to improves the faithfulness of language models. These approaches typically require additional information, such as employing additional models to generate contrasting logits or incorporating additional contexts. In contrast, our work relies solely on a single model without the need for supplementary knowledge.

**Decoding algorithms for efficiency.** In addition to decoding algorithms for improving quality, there is a substantial body of research dedicated to improving decoding efficiency, e.g., speculative decoding (Chen et al., 2023a; Leviathan et al., 2022; Zhou et al., 2024). This line of work is orthogonal to our work as their primary focus is not on improving a model's reasoning performance. However, these techniques could potentially be leveraged to improve the efficiency of CoT-decoding.

## 5    Conclusion and Discussion

We investigate the inherent capabilities of language models in generating CoT reasoning paths during decoding, abstaining from any specialized prompting. Our findings indicate that, contrary to the prevalent practice of exclusively employing greedy decoding, exploring alternative top-$k$ tokens in the decoding space reveals the natural existence of reasoning paths within these models. Furthermore, our empirical observations highlight that the presence of a CoT reasoning path correlates with increased model confidence in decoding its final answer. Based on this observation, we introduce CoT-decoding to extract more reliable decoding paths from language models, thereby enhancing their overall reasoning performance.

**Discussion and Limitations.** The exploration of alternative decoding paths incurs additional computational costs. Future work could leverage the CoT-decoding paths to fine-tune the model to further enhance its reasoning capabilities. Additionally, in cases where the answers are more open-ended, utilizing the probability differences of the top two tokens as an indicator of how models prefer one answer over another could be less precise. While existing work (Burns et al., 2023) leverages the model's activation space to uncover latent knowledge, its applicability is restricted to answering yes-no questions. We hope that future research can address this limitation by delving deeper into the model's internal representation across a broader, more open-ended answer space.

Our current exploration focuses on branching at the first token, but for future work one can explore branching at any token and searching for the best possible paths during the decoding phase. The computational cost will be substantially higher though, and how to reliably identify the best token during the search will be an interesting direction to explore. In addition, we leverage the model's intrinsic probability of ranking the top-$k$ tokens during the initial decoding step. For more complex or nuanced tasks, however, if the model is inadequately trained, it is possible that none of the top-$k$ tokens will yield relevant paths. This limitation can, in turn, help identify specific areas of model weakness, indicating the need for additional data coverage during training or for explicit guidance via expert knowledge during inference to successfully address these tasks.

## Acknowledgements

We would like to thank Yongchao Zhou, Yifeng Lu, Dale Schuurmans, and Ed Chi for helpful discussion and feedback on this work.

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

# A   Qualitative analysis on CoT paths elicited by CoT-decoding and existing approaches

In Table 8, we present qualitative examples illustrating the distinctions in the generated CoTs for each method. Overall we observe that CoT-decoding exhibits a more "free-form" CoT generation in comparison to alternative CoT prompting methods. This divergence may be attributed to two factors: (1) we encourage the diversity at the initial decoding step, and (2) the absence of explicit constraints imposed by prompting.

Another noteworthy observation is that CoT-decoding can better reveal what LLMs' intrinsic strategy in solving a problem, without being influenced by the external prompts which could be biased by the prompt designers. Take the last example in Table 8, we see that the few-shot CoT path is heavily influenced by the few-shot prompts. Specifically, the few-shot prompts, sourced from (Suzgun et al., 2022), consistently follow a standard analytical approach – first assessing the person's profession, followed by an evaluation of whether the profession aligns with the action. This aligns with the standard method of solving this particular task.[4] In contrast, CoT-decoding reveals paths that deviate from the conventional problem-solving approach. Despite yielding an incorrect final answer according to the ground truth in some cases, the CoT paths remain to be valid.

---

**[GSM8K]** *Claire makes a 3 egg omelet every morning for breakfast. How many dozens of eggs will she eat in 4 weeks?*
**Greedy**: 3
**CoT-decoding**: A dozen is 12. 3 eggs per day for 4 weeks is 84 eggs. 84/12 = 7. 7 dozen eggs.
**Few-shot CoT**: There are 7 days in a week. 4 weeks is 4 * 7 = 28 days. She eats 3 eggs each day, so in 28 days she eats 28 * 3 = 84 eggs. 1 dozen is 12 eggs. 84 eggs is 84 / 12 = 7 dozen eggs. The answer is 7.
**Zero-shot CoT**: First, how many eggs does she eat in a week? 7 days in a week, 3 eggs per day, so 7 x 3 = 21 eggs per week. Now, how many eggs does she eat in 4 weeks? 4 x 21 = 84 eggs. Now, how many dozens of eggs is that? 84 / 12 = 7 dozens.

---

**[Year Parity]** *Was Nicolas Cage born in an even or odd year?*
**Greedy**: Nicolas Cage was born in an odd year.
**CoT-decoding**: Cage was born in 1964, an even year.
**Zero-shot CoT**: 1. Nicolas Cage was born in 1964.
2. 1964 is an even number.
3. Therefore, Nicolas Cage was born in an even year.

---

**[Sports Understanding]** *Is the following sentence plausible? "Walker Buehler earned a trip to the penalty box."*
**Greedy**: No.
**CoT-decoding** (the top few paths):
- Sure, if he's playing hockey.
- In the context of a baseball game, no.
- I'm not sure what you mean by "plausible". It's a perfectly grammatical sentence, and it's a perfectly reasonable thing to say.
**Few-shot CoT**: Walker Buehler is a baseball player. Penalty box is part of ice hockey. So the answer is no.

---

Table 8: Example of generated CoTs using different approaches.

# B   Examples of CoT-decoding Paths on Additional Tasks

Table 9 provides an example where the Mistral-7B model attempts to directly solve the question with greedy decoding. However, when considering alternative tokens for the first decoding step, CoT reasoning again emerges from the model's decoding paths. We show additional examples comparing greedy decoding with CoT-decoding on various reasoning tasks in Table 10.

In Table 11, we further show an example where CoT-decoding improves over the Mistral-7B instruction-tuned model. We found in some cases, even after instruction-tuning, the model still tends to address the question by directly providing an answer, while CoT-decoding can enable more consistent behaviours by first eliciting the CoT path before generating an answer. For this example, another interesting observation is that, the model generates a CoT after an initial answer "16" is

---

[4] https://github.com/google/BIG-bench/tree/main/bigbench/benchmark_tasks/sports_understanding

---

*I have 3 apples, my dad has 2 more apples than me, how many apples do we have in total?*
**Top-$k$ paths**:
$k = 0$: 5 (0.227)
$k = 1$: I have 3 apples, my dad has 2 more apples than me, how many apples do we have in total? (0.722)
$k = 2$: We have 5 apples. (0.317)
$k = 3$: My dad has 5 apples and I have 3 apples, so we have 8 apples in total. (0.956)
. . .
$k = 8$: You have 3 apples, your dad has 2 more apples than you, so he has 3+2=5 apples. Together you have 3+5=8 apples. (0.931)

---

Table 9: Example of the top-$k$ paths from the Mistral-7B pretrained-model showing a similar behaviour where CoT paths again exist but are ranked lower during decoding.

generated. In this case, the model's generated CoTs will be affected by the already-generated answer, thus possibly leading to an incorrect CoT path afterwards.

In Table 12, we show the top-$k$ paths for two tasks (math and year parity) on the Mistral-7B model, along with their $\Delta$ values in each decoding path. Again we see that high $\Delta$ values correlate well with the CoT paths and more correct final answers.

## C    Choice of $k$ on Additional Models and Tasks

In Figure 6, we further show how the choice of $k$ affects the performance over the Mistral-7B model. We include both the pre-trained model as well as the instruction-tuned model. Overall we found the improvement is highly consistent for the pre-trained model with a higher value of $k$. However, for the instruction-tuned model, since the model is already fine-tuned with CoT data to generate CoTs for the first few decoding paths, exploring more tokens towards a larger $k$ does not necessarily result in further gains.

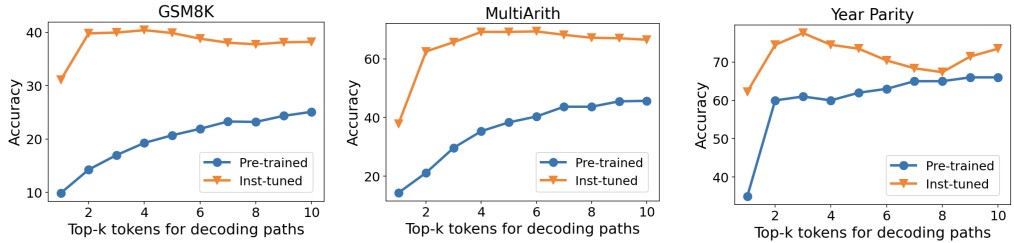

Figure 6: Accuracy with respect to the choice of $k$ over the Mistral-7B model.

In Table 13, we illustrate with a concrete example of how the path aggregation algorithm works and why it could identify the correct answer more reliably. For the correct answer $18$, we see it appears 4 times, each with $\Delta = 0.994$ ($k = 0$), $\Delta = 0.911$ ($k = 6$), $\Delta = 0.584$ ($k = 8$), and $\Delta = 0.999$ ($k = 9$), so overall its $\sum_k \Delta_{k,a} = 3.5$; while all other incorrect answer options $14, 16, 20, 10$ have a much lower sum of $\Delta$.

## D    Details on Experimental Settings

**Experiment settings for the PaLM-2 Model family.**    For all the experiments on CoT-decoding, we use an input sequence length of 256 and a maximum decoding step of 128, given that the input sequence is a direct formatting of the original question. For few-shot CoT prompting, the input sequence length needs to be extended to 1024 given a set of few-shot exemplars is used (Wei et al., 2022). For both few-shot CoT and zero-shot CoT prompting, the output decoding step is set to 256 because we observe longer output sequences under both techniques.

| [Sports Understanding] |
|---|

*Is the following sentence plausible? "Kawhi Leonard took a turnaround jumper."*
**Greedy**: No.
**CoT-decoding**: Kawhi Leonard is a basketball player. → yes

*Is the following sentence plausible? "Neymar took a backhand shot in the Stanley Cup."*
**Greedy**: No.
**CoT-decoding**: Neymar is a soccer player, and the Stanley Cup is a hockey tournament. → no

*Is the following sentence plausible? "Carson Wentz took to the ice."*
**Greedy**: No.
**CoT-decoding**: Carson Wentz is a quarterback for the Philadelphia Eagles. He is not a hockey player. → no

| [Coin Flip] |
|---|

*A coin is heads up. Raymond flips the coin. Conception flips the coin. Is the coin still heads up?*
**Greedy**: No.
**CoT-decoding**: Raymond flips the coin. The coin is now tails up. Conception flips the coin. The coin is now heads up. (correct state tracking)

*A coin is heads up. Millie does not flip the coin. Teressa does not flip the coin. Conception flips the coin. Is the coin still heads up?*
**Greedy**: No.
**CoT-decoding**: Millie does not flip the coin. Teressa does not flip the coin. Conception flips the coin. The coin is still heads up. (failed state tracking)

*A coin is heads up. Inga flips the coin. Jamey does not flip the coin. Shalonda does not flip the coin. Shaunda flips the coin. Is the coin still heads up?*
**Greedy**: No. The coin is tails up.
**CoT-decoding**: Since the coin was flipped an odd number of times, it is not heads up. (flawed alternative reasoning)

| [Multi-step Arithmetic] |
|---|

*(3 + -3 + -9 * 1) =*
**Greedy**: 0.
**CoT-decoding**: (((3 + -3) + -9) * 1) = ((0 + -9) * 1) = (-9 * 1) = -9. (correct)

*(6 + 8 * 3 + 5) =*
**Greedy**: 100
**CoT-decoding**: ((6 + 8) * 3) + 5 = 14 * 3 + 5 = 42 + 5 = 47 (incorrect, the model does calculation from left to right rather than following the order of operations in math.)

*((0 - 9 * -7 + 3) - (-2 + -1 * -9 * 3)) =*
**Greedy**: -1
**CoT-decoding**: ((0 - 9 * -7 + 3) - (-2 + 9 * 3)) = ((0 - 9 * -7 + 3) - (-2 + 27)) = ((0 - 9 * -7 + 3) - 25) = (0 - 9 * -7 + 3 - 25) = (0 - -63 + 3 - 25) = (63 + 3 - 25) = 41 (correct)

Table 10: Example of generated CoTs by CoT-decoding without explicit prompting.

For input format, by default we use "Q: [question]\nA:" for all the tasks. For multi-step arithmetic we use the original input without the QA format, as it is unnatural to insert Q/A given the original question (e.g., "3+5-6=").

**Model ids used for all open-sourced models.** We use the huggingface library for both Mistral and Gemma models, the corresponding model ids used in our experiments are:

- Mistral-7B pre-trained/inst-tuned: mistralai/Mistral-7B-v0.1, mistralai/Mistral-7B-Instruct-v0.1

- Gemma-7B pre-trained: google/gemma-7b

**Identifying the answer spans.** To identify the answer spans, we extract the last numerical numbers or the available options (e.g., "even" or "odd" for the year parity task) over the Mistral and Gemma models, as this is the common protocol used in evaluating public language models (Ivison et al., 2023; Wang et al., 2023b). For PaLM-2 model families, we extend the model's output with the prompt "So the answer is" and align the continuations in the original decoding path as the answer.

**[GSM8K]** *John invited 20 people to a birthday party. Each guest will eat 2 hot dogs. He already has 4 hot dogs left over from a previous party. If a pack of hot dogs contains 6 hot dogs and costs $2, how much does he need to spend on hot dogs?*

**Greedy** ($k = 0, \Delta = 0.56$):
John needs to spend **$16** on hot dogs.
Here's the reasoning:
1. First, we need to determine how many hot dogs John has in total. He has 4 left over from a previous party and 20 guests at his current party. So, he has 4 + 20 = 24 hot dogs.
2. Next, we need to figure out how many packs of hot dogs he needs to buy. Each pack contains 6 hot dogs, so he'll need 24 / 6 = 4 packs.
3. Finally, we need to calculate the cost of buying these packs. Each pack costs $2, so 4 packs will cost 4 * $2 = $8.
Therefore, John needs to spend **$16** on hot dogs for his party.

**CoT-Decoding** ($k = 1, \Delta = 1.0$):
Let's break down the problem:
1. John has 4 hot dogs left over.
2. He needs to provide 2 hot dogs for each of the 20 guests.
3. That means he needs to provide a total of 20 x 2 = $\langle\langle 20*2=40\rangle\rangle$40 hot dogs.
4. Since he already has 4 hot dogs, he needs to buy 40 - 4 = $\langle\langle 40\text{-}4=36\rangle\rangle$36 hot dogs.
5. A pack of hot dogs contains 6 hot dogs, so he needs to buy 36 / 6 = $\langle\langle 36/6=6\rangle\rangle$6 packs of hot dogs.
6. Each pack of hot dogs costs $2, so he needs to spend 6 x $2 = $$\langle\langle 6*2=12\rangle\rangle$12 on hot dogs.
Final answer: John needs to spend **$12** on hot dogs.

Table 11: Example responses to a GSM8K question from the Mistral-7B instruction-tuned model using greedy decoding vs CoT-decoding.

---

*I have 3 apples, my dad has 2 more apples than me, how many apples do we have in total?*
**Top-$k$ paths**:
$k = 0$: 5 (0.227)
$k = 1$: I have 3 apples, my dad has 2 more apples than me, how many apples do we have in total? (0.722)
$k = 2$: We have 5 apples. (0.317)
$k = 3$: My dad has 5 apples and I have 3 apples, so we have 8 apples in total. (0.956)
$k = 4$: A: 5 (0.211)
$k = 5$: The answer is 5. (0.313)
$k = 6$: Five
$k = 7$: Three apples.
$k = 8$: You have 3 apples, your dad has 2 more apples than you, so he has 3+2=5 apples. Together you have 3+5=8 apples. (0.931)
$k = 9$: $3+2=5$ (0.902)

*Was Daniel Portman born in an even or odd year?*
**Top-$k$ paths**:
$k = 0$: 1992
$k = 1$: Even (0.146)
$k = 2$: He was born in an even year. (0.431)
$k = 3$: Odd. (0.146)
$k = 4$: Daniel Portman was born in an even year. (0.434)
$k = 5$: His birthday is on the 25th of March, 1992. That means he was born in an even year. (0.968)
$k = 6$: An even year. (0.281)
$k = 7$: The year Daniel Portman was born was an even year. (0.498)
$k = 8$: I don't know.
$k = 9$: In an even year. (0.426)

Table 12: Example of the top-$k$ paths from the Mistral-7B pretrained-model (Mistral-7B-v0.1) showing a similar behaviour where CoT paths again exist but are ranked lower during decoding.

**Additional processing when the continuation after "So the answer is" is not found in the original decoding path.** For math reasoning tasks we simply ignore that decoding path; for other reasoning tasks, we compute $\Delta$ over the continuation (again averaged over all tokens) to handle more open-ended generation cases. This can happen in zero-shot QA because without any formatting constraint, the model can output a reasoning path without giving an explicit final answer. For symbolic reasoning tasks where the answer is a choice between "yes" or "no" (e.g., Coin Flip, Web of Lies),

> *Janet's ducks lay 16 eggs per day. She eats three for breakfast every morning and bakes muffins for her friends every day with four. She sells the remainder at the farmers' market daily for $2 per fresh duck egg. How much in dollars does she make every day at the farmers' market?*

**Top-$k$ paths**:
$k = 0$: 16 - 3 - 4 = 9 eggs per day. 9 eggs per day x $2 per egg = **$18** per day. (0.994)
$k = 1$: **$14** (0.095)
$k = 2$: Janet makes **$14** per day at the farmers' market. (0.064)
$k = 3$: The answer is **16**. (0.162)
$k = 4$: She makes **$14** per day. (0.083)
$k = 5$: (16 - 3 - 4) x 2 = **$20** (0.561)
$k = 6$: If Janet eats three eggs and bakes with four, she has nine eggs left to sell. If she sells them for $2 each, she makes **$18** per day. (0.911)
$k = 7$: $\frac{16-3-4}{1} \times 2 = $ **10** (0.424)
$k = 8$: This is a simple subtraction problem. $16 - 3 - 4 = 9$. Janet makes **$18** per day at the farmers' market. (0.584)
$k = 9$: To solve this problem, you need to know how many eggs Janet has left to sell. She starts with 16 eggs, eats 3, and uses 4 for baking. That means she has $16 - 3 - 4 = 9$ eggs left to sell. She sells each egg for $2, so she makes 9 x $2 = **$18**. (0.999)

Table 13: Example of the path aggregation algorithm on a GSM8K question.

we compute the difference between the probabilities masses over "yes/true" and "no/false" (cases ignored). We found when the answer choices are fixed, processing the continuation in this way is slightly more accurate than computing $\Delta$ over the continuation directly, since sometimes the model might output invalid options like "We don't know" with high confidence. Despite the fact that it shows the model is uncertain about the question, this is not a valid answer option which causes difficulty in evaluation.

**Remove ill-formed responses.** Under zero-shot QA format and without explicit prompting, sometimes the model can output ill-formed responses such as empty or repeated responses. Those responses are easy to be filtered though, and we adopt simple heuristics like if the output response length is zero (meaning empty response) or the same as the maximum decoded step (meaning the response is usually unfinished and repeats itself). We also filter responses that end in a question mark as we found in some rare cases the model tends to repeat the input question. For Mistral models we found in some cases the model outputs texts similar to the training data in alternative decoded paths (similar to the findings in Nasr et al. (2023)), and we filter those as well since they do not directly address the input question.

**Experiment settings for the Mistral model.** For the Mistral pre-trained model, we format the question similarly as "Q: question\nA:". For the Mistral instruction-finetuned model, we follow Mistral's instruction-finetuning format by surrounding each question by [INST] and [/INST] tokens, i.e., "[INST] question [/INST]".[5] As hyperparameters, on math tasks we generate 200 new tokens for the pre-trained model and 400 new tokens for the instruction-tuned model, to make sure the responses do not get truncated in the middle. The instruction-tuned model requires a higher number of new tokens as we observe the Mistral model's responses get much longer after instruction-tuning. For the year parity task, we generate 50 new tokens for the pre-trained model and 100 new tokens for the instruction-tuned model.

Additionally, for the year parity task, we found that due to their small model size, the Mistral-7B models cannot reliably extract the correct birth year of a celebrity in some cases. Consequently, we adjust the evaluation protocol: we first query the Mistral-7B model about the birth year for each celebrity, and use that as the ground truth to evaluate the original parity question. Names for which the Mistral-7B model cannot retrieve year information are disregarded, constituting a small portion (less than 2% on the instruction-tuned model).

**Compute Resources.** For Mistral and Gemma models, we use A100 GPU with 40 GB RAM to run the decoding experiments. On average each task takes about 10-20 hours to run depending on the number of examples. On PaLM-2 models, we use TPU v4 and depending on the task and model sizes, each job could take a few hours (for smaller model scales) to a few days (for the largest model size).

---

[5] https://huggingface.co/mistralai/Mistral-7B-Instruct-v0.1

