# OpenReview forum: "Chain-of-Thought Reasoning Without Prompting"
_NeurIPS.cc/2024/Conference — NeurIPS 2024 poster_

### Official Review · Reviewer_Es5x · 2024-06-28

**Soundness:** 2
**Presentation:** 3
**Contribution:** 2
**Rating:** 4
**Confidence:** 4

**Summary:**

The paper investigates the inherent capabilities of LLMs to generate CoT reasoning paths. This study introduces CoT-decoding, a method that explores alternative top-k tokens in the decoding space to uncover reasoning paths without any specialized prompting. The findings indicate that the presence of a CoT reasoning path correlates with increased model confidence in its final answer. The paper highlights that CoT-decoding can enhance the reasoning performance of language models by extracting more reliable decoding paths.

**Strengths:**

1. The paper introduces CoT-decoding to elicit reasoning paths and provides an effective alternative to prompt engineering.
2. The results demonstrate significant improvements over greedy decoding, showcasing the practical benefits of the proposed method.
3. The study finds a clear correlation between the presence of CoT paths and increased model confidence, providing a useful metric for identifying reliable decoding paths.

**Weaknesses:**

1. The primary contribution of the paper appears to be a method for selecting a decoding path from the top-k generated paths. While this approach is useful, it lacks novelty and significant impact.
2. The motivation behind the study is unclear to me for several reasons. Firstly, providing questions alone constitutes a form of prompting instead of no prompting. Secondly, the use of a prompt that inherently challenges the elicitation of CoT reasoning paths via greedy decoding, only to then introduce a specialized decoding method to address this, strikes me as contrived. Lastly, I question the claim that CoT-decoding effectively enhances the reasoning capabilities of language models, as it merely selects from pre-generated paths rather than uncovering new, substantive reasoning processes.
3. The proposed method may face challenges in tasks where identifying correct answer spans is difficult, limiting its practical applicability.
4. The comparison between CoT-decoding and greedy decoding is unfair. CoT-decoding inherently requires multiple reasoning paths to be generated first, which is not a requirement for greedy decoding. The exploration of multiple decoding paths significantly increases the computational cost, which could be a drawback for practical applications.
5. The evidence provided in the paper is restricted to a few reasoning benchmarks and toy tasks. To convincingly demonstrate its utility, the method should be tested across a broader array of datasets, such as the MATH benchmark, and in more complex real-world scenarios.
6. The range of models evaluated in the study is too narrow. Including mainstream models like GPT-3.5, GPT-4, and diverse open-source models like llama3 in various sizes would provide a more robust evaluation of the method's effectiveness.

**Questions:**

NA

**Limitations:**

Yes

---

> ### Author Rebuttal · Authors · 2024-08-06
>
> Thank you for your constructive feedback!
>
> > The primary contribution of the paper appears to be a method for selecting a decoding path from the top-k generated paths. While this approach is useful, it lacks novelty and significant impact.
>
> To clarify our contribution, our first finding is that LLMs inherently possess reasoning capabilities even without any additional prompting, this is a *novel* finding in contrast to many previous papers proposing better prompting to enable LLM reasoning. Our other contribution is to propose CoT-decoding that identifies the correct CoT-path, this is also the *first decoding-only method* that effectively enables language models to reason, please see Table 4 for a detailed comparison to all existing decoding algorithms.
>
>
> > The motivation behind the study is unclear to me for several reasons. Firstly, providing questions alone constitutes a form of prompting instead of no prompting. Secondly, the use of a prompt that inherently challenges the elicitation of CoT reasoning paths via greedy decoding, only to then introduce a specialized decoding method to address this, strikes me as contrived. Lastly, I question the claim that CoT-decoding effectively enhances the reasoning capabilities of language models, as it merely selects from pre-generated paths rather than uncovering new, substantive reasoning processes.
>
> Thanks for the suggestion, we will add more clarification to the motivation as below.
>
> First, to clarify the “prompting” part: the question has to be an input to the model, otherwise the model will have no input :) In existing literature, the “prompting” part usually refers to any additional prompts added to the question, e.g., zero-shot CoT uses “let’s think step by step” and few-shot CoT uses few-shot demonstrations before the question. We will make this point more clear in the paper.
>
> Second, your comment is similar to what existing papers try to claim: LLMs *can’t reason* with questions only. Our study is the first to show that it is *not the case*: the observation that models can’t reason without additional prompting is due to the prevalent usage of greedy decoding, while the top-k alternative decoding paths can unveil the existence of CoTs.
>
> To your last point, yes as we emphasized in multiple places in our paper, our method “enables a better understanding of LLMs’ intrinsic reasoning capabilities” (line 60). Our primary finding is that LLMs can already reason by themselves after pre-training, and “the reasoning process can be readily elicited by simple decoding changes” (line 56).
> We say that the model’s reasoning performance is enhanced by CoT-decoding *when compared to greedy decoding*, and we do not claim that we enhance the inherent reasoning capabilities of LLMs, we just made those capabilities more prominent. We will make this point more clear in our paper.
>
> > The proposed method may face challenges in tasks where identifying correct answer spans is difficult, limiting its practical applicability.
>
> Yes, as we discussed in our limitation section, we hope this can be better addressed in future research by better learning the model’s internal representation across a broader, more open-ended answer space.
>
> > The comparison between CoT-decoding and greedy decoding is unfair. CoT-decoding inherently requires multiple reasoning paths to be generated first, which is not a requirement for greedy decoding. The exploration of multiple decoding paths significantly increases the computational cost, which could be a drawback for practical applications.
>
> Thanks for your question. We explore more decoding paths because we want to investigate whether the model inherently possesses the reasoning capability or not, which is currently masked due to the prevalent usage of greedy decoding. And yes as discussed in our limitation section, our method will incur more computational cost, and we can use the CoT paths found by our method to further train the model, such that those reasoning paths can be readily output during inference.
>
> > The evidence provided in the paper is restricted to a few reasoning benchmarks and toy tasks. To convincingly demonstrate its utility, the method should be tested across a broader array of datasets, such as the MATH benchmark, and in more complex real-world scenarios.
> The range of models evaluated in the study is too narrow. Including mainstream models like GPT-3.5, GPT-4, and diverse open-source models like llama3 in various sizes would provide a more robust evaluation of the method's effectiveness.
>
> Thanks for the suggestion. We include results on Llama-3 below, showing the robustness of our proposed approach. We also add the MATH benchmark (the MATH-500 held-out set from https://arxiv.org/pdf/2305.20050) to cover a broader range of datasets.
> For GPT models, note that the focus of our paper is on *pre-trained* models to investigate their intrinsic reasoning capabilities, while current exposed GPT APIs are all instruction fine-tuned models, hence it is hard to distinguish whether the model can reason already after pre-training, or acquired such ability during instruction fine-tuning via a substantial amount of CoT reasoning data. Also note that in Figure 3, we already plotted the performance of CoT-decoding on models with *various sizes* (XS, Small, Medium, Large) and showed performance improvement across the board.
>
> Results on Llama-3-8B pre-trained model:
>
> | | GSM8K | MultiArith | MATH |
> |---| --- | --- | --- |
> | greedy decoding | 21.4% | 41.8% | 14.0% |
> | CoT-decoding (max) | 32.7% | 50.7% | 19.8% |
> | CoT-decoding (agg) | 47.9% | 77.8% | 22.6% |

---

### Official Review · Reviewer_C32n · 2024-07-10

**Soundness:** 3
**Presentation:** 3
**Contribution:** 2
**Rating:** 6
**Confidence:** 4

**Summary:**

The paper explores the inherent reasoning capabilities of large language models (LLMs) without the need for explicit prompting. By altering the decoding process to consider top-k alternative tokens, the authors reveal that Chain-of-Thought (CoT) reasoning paths can emerge naturally. This approach bypasses the need for task-specific prompt engineering and demonstrates that LLMs possess intrinsic reasoning abilities. Extensive empirical evaluations across various reasoning benchmarks show significant performance improvements using CoT-decoding compared to conventional greedy decoding.

**Strengths:**

- The paper introduces a novel approach to reveal LLMs' reasoning capabilities without explicit prompts by altering the decoding process.
- The paper is clearly written, with detailed explanations and illustrations of the CoT-decoding process.
- The findings challenge the prevailing notion that LLMs require specific prompting for reasoning, highlighting the models' intrinsic abilities.

**Weaknesses:**

- The approach, while novel in its specific application, builds on existing concepts of decoding and model confidence. The novelty is somewhat incremental.
- The scope of experiments could be broadened to include more diverse and complex reasoning tasks. Additional benchmarks and comparisons with state-of-the-art prompting methods would strengthen the paper.
- The paper could benefit from a more detailed comparative analysis with other decoding and prompting strategies to contextualize its contributions better.

**Questions:**

- Can the authors provide more detailed comparisons with other state-of-the-art prompting and decoding methods?
- How does the CoT-decoding method perform on more complex and diverse reasoning tasks not covered in the current experiments?
- Are there any potential limitations or biases introduced by relying on top-k alternative tokens for decoding?

**Limitations:**

The authors have addressed some limitations of their work, such as the additional computational costs incurred by exploring alternative decoding paths. However, the paper could further discuss the potential biases introduced by this approach and its applicability to more complex, open-ended tasks.

---

> ### Author Rebuttal · Authors · 2024-08-06
>
> Thank you for the feedback!
>
> > Can the authors provide more detailed comparisons with other state-of-the-art prompting and decoding methods? The paper could benefit from a more detailed comparative analysis with other decoding and prompting strategies to contextualize its contributions better.
>
> - Comparison to other decoding methods: Please see Table 4 for the detailed comparison on all popular decoding methods used by SoTA LLMs, including greedy, top-k, temperature sampling, nucleus sampling, beam search, and SoTA decoding methods like self-consistency (no prompt), and we show that CoT-decoding is the **only decoding algorithm** that significantly enhances reasoning in language models.
> - Comparison to other prompting methods: please see Table 7 for the results with CoT and self-consistency (with CoT prompt), where both are standard approaches used in major LLM reports to achieve SoTA.
> Note that CoT-decoding is a decoding algorithm which is **orthogonal** to existing prompting techniques, and (1) CoT-decoding can be easily combined with existing prompting techniques to yield further gains (see Table 7 for the results); (2) our paper aims to show that LLMs inherently possess reasoning abilities *without prompting*, hence adding more sophisticated prompting techniques is not the primary focus of this paper.
> - To contextualize our contributions better: existing literature proposing more complex prompting methods often requires adding task-specific human-prior and performing manually-intensive prompt-engineering. As a result, those methods may achieve better task-specific improvements but could be hard to scale or transfer poorly across tasks or models. In contrast, our method requires *no prompt-engineering*, *no human intervention*, is completely *task and model-agnostic* and improves reasoning across the board.
>
> > How does the CoT-decoding method perform on more complex and diverse reasoning tasks not covered in the current experiments?
>
> Our current experiments covering math, commonsense, and symbolic reasoning (with various difficulty levels) showed a *consistent trend*: CoT-decoding can effectively uncover a model's intrinsic reasoning capabilities previously obscured by greedy decoding, although the model's own reasoning ability varies depending on the task difficulty level, which can be attributed to the task prominence in its pre-training distribution.
> For example, in Figure 4, we show that when task difficulty increases or becomes more synthetic, it becomes harder to find the correct CoT paths (like tasks that require > 3 steps of accurate state tracking). Hence for a new task, the existence of CoT paths will depend on how prominent similar data exists in pre-training, and if relevant knowledge can be retrieved and the solution can be found within a few steps of knowledge manipulation.
>
> Below we also add results of the Llama-3-8B pre-trained model on the MATH benchmark, which is a substantially more diverse and complex reasoning dataset. We can see that CoT-decoding can still yield effective improvements over greedy decoding.
>
> |  | MATH accuracy |
> | -------- | ------- |
> | greedy decoding | 14.0% |
> | CoT-decoding (max) | 19.8% |
> | CoT-decoding (agg) | 22.6% |
>
>
> > Are there any potential limitations or biases introduced by relying on top-k alternative tokens for decoding?
>
> Thanks for the question. We rely on the model’s own top-k token ranking because we want to better understand the model's intrinsic reasoning capabilities. As our study shows in Section 3.2 and Figure 4, our analysis unveils the model's intrinsic vulnerabilities in reasoning, e.g., on Big-bench tasks with controlled-difficulty levels, the model's ranking accuracy of top-k tokens becomes lower as the task complexity increases.
> Hence for highly difficult tasks, if the model is not well-trained, it’s possible that none of the top-k tokens yield relevant paths. This in turn can help us identify a model's areas of weakness though, so we know that we need additional data coverage in model training, or we need to explicitly guide the model via expert knowledge during inference to solve those tasks.
>
> > The approach, while novel in its specific application, builds on existing concepts of decoding and model confidence. The novelty is somewhat incremental.
>
> Note that the detail of CoT-decoding differs substantially from existing top-k decoding or model confidence estimation, hence CoT-decoding is novel in its design and effectiveness in significantly improving LLM reasoning:
> - Existing top-k decoding is a sampling algorithm, where each token is *sampled* to enhance the diversity in the *overall sequence*. While in CoT-decoding, other than the first token, all remaining tokens are greedily generated. The reason is that we want to encourage diversity in the 1st token such that the model can escape from the local optima, while for the rest of the tokens we want the model to follow the optimal reasoning path hence they’re still greedily generated. This is also different from beam search because beam search uses the model’s sequence probability to rank the responses. In contrast, in CoT-decoding we observe that the model's sequence probability is not reliable for reasoning tasks (Table 2), while the model's final answer confidence score proves to be significantly more accurate in identifying the correct CoT paths.
> - Model confidence estimation: note that our confidence estimation is based on our novel observation that responses with a CoT first typically have a more confidently-decoded final answer. This is also very different from existing works that try to estimate the model’s confidence for the whole sequence to identify scenarios where the model is uncertain.

---

> > ### Comment · Reviewer_C32n · 2024-08-10
> >
> > Thanks for your rebuttal, I will keep my positive ratings.

---

> > > ### Author Response · Authors · 2024-08-13
> > >
> > > Thank you for acknowledging our rebuttal. We will add these additional discussion points to the paper.

---

### Official Review · Reviewer_BrnW · 2024-07-13

**Soundness:** 3
**Presentation:** 3
**Contribution:** 3
**Rating:** 8
**Confidence:** 5

**Summary:**

The paper investigates an innovative approach to eliciting chain-of-thought (CoT) reasoning from pre-trained large language models (LLMs) without the need for explicit prompting techniques, which typically require intensive prompt engineering and can obscure the model's inherent reasoning capabilities. Instead, this study proposes altering the decoding process by exploring alternative top-k tokens, rather than the conventional top-1 greedy decoding path. This method, termed CoT-decoding, effectively reveals natural CoT reasoning paths within the language model's outputs, leading to more accurate and confident model responses across various reasoning tasks.
Contributions:
1. **Novel Decoding Strategy:** The paper introduces a novel decoding strategy that bypasses the need for prompt engineering by modifying the decoding process to explore alternative token paths, allowing the model to display its intrinsic reasoning capabilities.
2. **Empirical Validation:** Extensive empirical studies demonstrate that this CoT-decoding approach not only enhances the accuracy of LLMs on reasoning tasks but also increases the confidence of the models in their answers, suggesting a higher reliability of the reasoning paths discovered.
3. **Comparative Analysis:** The paper provides a comparative analysis of CoT-decoding against traditional methods, showing significant improvements over greedy decoding and other baseline methods across multiple benchmarks for mathematical and commonsense reasoning tasks.

**Strengths:**

1. The paper introduces a novel approach to eliciting chain-of-thought (CoT) reasoning from pre-trained large language models (LLMs) without requiring explicit prompting. This method diverges from conventional prompting techniques, which involve either few-shot or zero-shot prompting, by modifying the decoding process to explore alternative token paths. This strategy is both innovative and creative as it challenges the standard practice of prompt engineering and demonstrates that CoT reasoning can be naturally elicited through decoding adjustments.
2. The paper is supported by extensive empirical studies that validate the efficacy of the CoT-decoding method. The experiments are well-designed, covering a broad range of reasoning tasks, including mathematical and commonsense reasoning benchmarks.
3. The paper is exceptionally clear and well-organized. The authors provide a detailed description of the CoT-decoding process, accompanied by illustrative figures and examples that help clarify how the method diverges from traditional decoding strategies.

**Weaknesses:**

1. The method involves generating multiple decoding paths and evaluating them to identify the most confident reasoning trajectory, which could be computationally expensive, especially when applied at scale or in real-time applications.
2. The approach heavily relies on the selection of top-k alternative tokens, which may not always yield the most relevant or coherent paths for reasoning, especially in more complex or nuanced tasks.

**Questions:**

See weekness.

**Limitations:**

None. Good Paper~

---

> ### Author Rebuttal · Authors · 2024-08-06
>
> Thank you for the thoughtful review!
>
> > The method involves generating multiple decoding paths and evaluating them to identify the most confident reasoning trajectory, which could be computationally expensive, especially when applied at scale or in real-time applications.
>
> Thanks for the feedback. Yes as we discussed in the limitation section, our method does incur higher computational cost when we try to identify those CoT-paths. For real-time applications, we can incorporate the “good paths” found by our CoT-decoding algorithm into model training, such that during inference the model can directly output those good paths as the top-paths.
> For the study in this paper, we spent higher compute to get the top-k paths mainly to investigate 1) whether CoT-paths exist in the top-k paths, and 2) how large k needs to be for us to find a CoT-path. Figure 4 shows that for many tasks, simply increasing k>1 can already help uncover many of the previously-hidden CoT paths.
>
> > The approach heavily relies on the selection of top-k alternative tokens, which may not always yield the most relevant or coherent paths for reasoning, especially in more complex or nuanced tasks.
>
> Thanks for this question. Yes, we rely on the model's own probability of ranking the top-k tokens during the first decoding step, and we do observe that for highly difficult tasks, if the model is not well-trained, it’s possible that none of the top-k tokens yields relevant paths. This in turn can help us identify a model's areas of weakness though, so we know that we need additional data coverage in model training, or we need to explicitly guide the model via expert knowledge during inference to solve those tasks. We will add more discussion on this point to make it more clear.

---

> > ### Comment · Reviewer_BrnW · 2024-08-09
> > **Official Comment by Reviewer BrnW**
> >
> > Thank you for your thorough rebuttal and for addressing the concerns raised. I believe this work is very meaningful and provides valuable insights into enhancing the model's reasoning capabilities and constructing higher-quality datasets. In light of your detailed response and the efforts made to address these complexities, I have decided to increase my score from 7 to 8.

---

> > > ### Author Response · Authors · 2024-08-09
> > >
> > > Thank you for carefully reading through our rebuttal and raising your score. We will add these additional discussion points into our paper. We truly appreciate your time in reviewing this work.

---

### Official Review · Reviewer_fBkE · 2024-07-15

**Soundness:** 3
**Presentation:** 3
**Contribution:** 3
**Rating:** 8
**Confidence:** 4

**Summary:**

The paper investigates the intrinsic reasoning capabilities of LLMs without relying on prompting techniques like few-shot or zero-shot prompting. The study introduces an alternative approach by altering the decoding process, specifically by exploring the top-k alternative tokens rather than following the standard greedy decoding path. The proposed "CoT-decoding," reveals that LLMs inherently possess chain-of-thought reasoning abilities that are often hidden by human-involved prompting / conventional decoding methods.

**Strengths:**

- The paper demonstrates that LLMs can generate CoT reasoning paths without explicit prompts by modifying the decoding process. This challenges the prevailing notion that LLMs require prompting to reason effectively. It shows that LLMs have inherent reasoning capabilities that are more accurately assessed by CoT-decoding.
- Experimental results show that CoT-decoding naturally reveals CoT reasoning paths during the decoding process, significantly enhancing the model's reasoning capabilities and surpassing greedy decoding. It is also observed that these paths are more prevalent in tasks frequently encountered in pre-training data.
- The paper is generally well-written and easy to follow.

**Weaknesses:**

- **Extra Computational Cost**: Exploring top-k alternative tokens for decoding paths requires additional computational resources. Compared to CoT/Zero-shot CoT/ComplexCoT/CoT-SC(n) where n is not a large number, CoT-decoding necessitates evaluating multiple decoding paths, which increases computational complexity and time cost, especially when dealing with complex multi-step tasks.
- **Limitations in Open-ended Questions**: This method mainly relies on the model's confidence in specific answers during the decoding process to select reasoning paths. However, for more open-ended questions, this probability difference-based selection method may not be precise enough, making it challenging for LLMs to select the optimal path in a much larger answer space.

**Questions:**

Does the limitation of branching only at early decoding stages restrict the flexibility and applicability of CoT-decoding? Would exploring branching at later stages in the decoding process improve overall performance?

**Limitations:**

I do not identify any negative societal impacts in this paper.

---

> ### Author Rebuttal · Authors · 2024-08-06
>
> Thank you for your thoughtful review!
>
> > Does the limitation of branching only at early decoding stages restrict the flexibility and applicability of CoT-decoding? Would exploring branching at later stages in the decoding process improve overall performance?
>
> Thanks for the question. We do have a study of branching at later stages in Figure 6, Appendix D due to space limit, and we show that branching at later steps is viable but incurs much higher computational cost.
> For most tasks, we found that early branching significantly improves the diversity of potential paths, and is usually sufficient to uncover the CoT paths. With later branching, sometimes the model encounters difficulty recovering from incorrect paths (e.g., for the math question, after generating the token "5" the model is not able to recover from an erroneous path).
> For some tasks though (e.g., the year parity task), mid-branching can help uncover more diverse paths, potentially leading to a better performance. We think it would be an interesting future direction to determine the optimal branching points depending on the specific task/question and efficiently search for the best possible paths.
>
> > Extra computational cost and limitation to open-ended questions.
>
> Thanks for the feedback. Yes, as discussed in the limitation section, our method does incur higher computational costs and in practice, we can use the optimal paths identified from CoT-decoding to further fine-tune the language model, which can help the model directly output those paths during inference time.
>
> For open-ended questions, we discussed that for some tasks identifying the answer could be non-trivial, and we hope this can be better addressed in future research by better learning the model’s internal representation across a broader, more open-ended answer space. Fine-tuning with CoT-decoding paths can potentially help here as well, as the model learns to output its thinking process first when it is uncertain on open-ended questions.

---

> ### Comment · Reviewer_fBkE · 2024-08-13
> **Thanks for your rebuttal**
>
> Thank you very much for the rebuttal. I found the information I wanted in the appendix. CoT-Decoding is an interesting discovery, and I have raised my score from 7 to 8, leaning toward clear acceptance.

---

> > ### Author Response · Authors · 2024-08-13
> >
> > Thank you for going through our rebuttal and raising the score. We will move the branching discussion to the main text to make this information more accessible.

---

### Decision · Program_Chairs · 2024-09-25

**Decision:**

Accept (poster)

**Comment:**

This paper investigates whether LLMs can generate CoT paths without any prompting (i.e., only input a question to the model, without additional info).This is in contrast to prior work that focuses on developing different prompts (e.g., few-shot prompts, “let’s think step by step”) to elicit reasoning steps from LLMs. The findings suggest that CoT reasoning paths can be elicited from pre-trained LLMs by simply considering the alternative paths greedily decoded from the top-k tokens at decoding step 0. The authors have added a few more experiments during the rebuttal, which make the results more convincing. Most reviewers tend to agree that the paper introduces a novel approach, very well-written and easy to follow. I would recommend accepting this paper and the authors follow the suggestions and discussions during rebuttal to refine it.